# Deep-learning-based segmentation using individual patient data on prostate cancer radiation therapy

**Sangwoon Jeong**[1], **Wonjoong Cheon**[2], **Sungjin Kim**[3], **Won Park**[4], **Youngyih Han**[1,4]*

**1** Department of Health Sciences and Technology, SAIHST, Sungkyunkwan University, Seoul, Korea,
**2** Department of Radiation Oncology, Seoul St. Mary's Hospital, College of Medicine, The Catholic University of Korea, Seoul, Korea, **3** Department of Radiation Oncology, Samsung Medical Center, Seoul, Korea,
**4** Department of Radiation Oncology, Samsung Medical Center, Sungkyunkwan University School of Medicine, Seoul, Korea

* youngyih@skku.edu

**Data Availability Statement:** the code file will be available in the following address. https://github.com/SangWoonJeong/DVF-based-segmentation.

## Abstract

### Purpose

Organ-at-risk segmentation is essential in adaptive radiotherapy (ART). Learning-based automatic segmentation can reduce committed labor and accelerate the ART process. In this study, an auto-segmentation model was developed by employing individual patient datasets and a deep-learning-based augmentation method for tailoring radiation therapy according to the changes in the target and organ of interest in patients with prostate cancer.

### Methods

Two computed tomography (CT) datasets with well-defined labels, including contoured prostate, bladder, and rectum, were obtained from 18 patients. The labels of the CT images captured during radiation therapy ($CT^{2nd}$) were predicted using CT images scanned before radiation therapy ($CT^{1st}$). From the deformable vector fields (DVFs) created by using the VoxelMorph method, 10 DVFs were extracted when each of the modified CT and $CT^{2nd}$ images were deformed and registered to the fixed $CT^{1st}$ image. Augmented images were acquired by utilizing 110 extracted DVFs and spatially transforming the $CT^{1st}$ images and labels. An nnU-net autosegmentation network was trained by using the augmented images, and the $CT^{2nd}$ label was predicted. A patient-specific model was created for 18 patients, and the performances of the individual models were evaluated. The results were evaluated by employing the Dice similarity coefficient (DSC), average Hausdorff distance, and mean surface distance. The accuracy of the proposed model was compared with those of models trained with large datasets.

### Results

Patient-specific models were developed successfully. For the proposed method, the DSC values of the actual and predicted labels for the bladder, prostate, and rectum were 0.94 ± 0.03, 0.84 ± 0.07, and 0.83 ± 0.04, respectively.

**Funding:** @ Initials of the authors who received each award: Y.Han @ Grant numbers: 2022R1A2C2004694 and 2021M2E8A1048108 @The full name of each funder: National Research Foundation of Korea grants funded by the Korean government (MSIT) @URL of each funder website: https://www.nrf.re.kr/eng/index @Did the sponsors or funders play any role in the study design, data collection and analysis, decision to publish, or preparation of the manuscript?: No.

**Competing interests:** NO authors have competing interests.

## Conclusion

We demonstrated the feasibility of automatic segmentation by employing individual patient datasets and image augmentation techniques. The proposed method has potential for clinical application in automatic prostate segmentation for ART.

## 1. Introduction

Diseases related to the prostate are common in adult males, and prostate cancer is the second leading cause of cancer-related deaths in the United States (the first is lung cancer) [1]. In addition, incidences of prostate cancer are increasing in Asia and Europe and are expected to increase globally, along with mortality rates [2]. Prostatectomy and radiation therapy are the common treatment options for clinically localized prostate cancer [3]. Radiation therapy is a noninvasive treatment that destroys cancer cells via irradiation with high-energy radiation [4,5]. Radiation therapy accurately depicts the target and the organ at risk (OAR) in computed tomography (CT) images; the highest possible dose is provided to the target while irradiating the OAR with the minimum dose through the established treatment plan. As radiation therapy is conducted daily for four to eight weeks, the volume and shape of the target and OAR can change [6] owing to changes in the patient's physiology, such as weight loss, movement of body fluids, and tumor volume reduction [7]. Woodford et al. [8] reported that the gross tumor volume during 30 fractions was reduced by an average of 38%, ranging from 12% to 87%. Romero et al. [9] demonstrated a maximum dose loss of 21.1% for the tumor volume when treatment was continued with the initially established treatment plan. Failure to adapt the treatment to changes in the target or OAR can result in treatment failure or radiation overdose of up to 36.8% in the OAR, which can cause unwanted acute or late toxicity [10]. Therefore, radiation treatment strategies considering these anatomical changes are required.

Adaptive radiation therapy (ART) is a radiation treatment strategy in which the detected anatomical changes in a patient are considered and a modified treatment plan established during the treatment course [11]. ART enables the accurate treatment of cancer. However, re-establishing treatment plans and recontouring the target and OAR for each modification require additional resources. In particular, OAR recontouring is a time-consuming process that requires approximately 180 min [12]. In addition, the contouring technology employed by dosimetrists has considerable impact on the consistency and accuracy of OAR segmentation [13].

Automated segmentation techniques have been proposed to reduce committed labor and accelerate the ART process. Schulz et al. [14] developed a semiautomatic contouring method that demonstrated a 30% reduction in the time required for OAR segmentation compared with manual contouring. Doshi et al. [15] proved the usability of semiautomatic contouring, wherein the average Dice similarity coefficient (DSC) between semiautomatic and manually segmented structures was 87% in tongue tumors. Although the contouring time can be reduced by employing semiautomatic segmentation, the segmented OAR must be manually modified owing to the incompleteness of automation [16]. Recently, automatic segmentation techniques that do not require additional manual modifications have been developed to enhance segmentation accuracy.

The first automatic image segmentation methods to be introduced included edge-detection-based [17], threshold-based [18], and region-based image segmentation methods [19]. With the recent developments in deep learning, a new generation of automatic segmentation

methods, such as DeepLab v1 [20], DeepLab v2 [21], and SegNet [22], has been proposed to utilize fully convolutional neural network (FCN)-based up-sampling. In particular, since U-net [23], which employs FCN-based up- and down-sampling, was introduced, various modifications have been made to the network structure to improve the segmentation performance [24,25]. In particular, automatic segmentation methods utilizing a generative adversarial network (GAN), such as the GAN-based SegGAN [26], SegmaticGAN [27], and Unet-GAN [28], have improved the segmentation performance [29,30].

Developing a high-performance network requires large amounts of data for network training. However, acquiring many medical images is difficult because of patient privacy and security issues [31]. Even if the medical images are acquired, creating a manually segmented OAR, which is termed as a "label," is time-consuming and expensive [32]. Employing pretrained deep-learning models can reduce the required amount of image data for training; however, a model's architecture can require different modalities and image sizes, limiting the usability of the models and reducing the prediction performance [33,34]. To overcome these restrictions, an image augmentation method that modifies the original image data has been proposed to increase the number of datasets [35]. Image augmentation methods can be categorized into rigid and nonrigid methods. The rigid method produces geometric variations in the original image but has limitations in creating human body images of various shapes; whereas the nonrigid method can create human body images having various shapes but can occasionally create unrealistic images [36]. Recently, augmentation employing a deep-learning method was proposed to create modified images that were similar to actual patient images [37]. Here, we investigated the feasibility of an automatic segmentation network that utilized only individual patient datasets. As the internal structure of each patient has its own characteristics and the deviation or change from the initial shape and position can have a limited range, we assumed that an individual model utilizing each patient's own data may have high performance. Thus, this study proposes an automatic segmentation model that employs an individual patient dataset and utilizes an image augmentation technique for tailoring radiation therapy according to the changes in the target and organ of interest in patients with prostate cancer.

For radiation therapy, a CT simulation must be performed ($CT^{1st}$) and a label must be generated in the process of creating a treatment plan. After the segmentation model was trained by using the $CT^{1st}$ image with the label and other augmented $CT^{1st}$ images, automatic segmentation was performed on the second CT image ($CT^{2nd}$) during ART. Nonrigid augmentation utilizing deep learning was employed to create the training datasets. Our assumption was validated by quantifying the model performance against the $CT^{2nd}$ label, which was manually contoured.

## 2. Materials and methods

The $CT^{1st}$ image and label and $CT^{2nd}$ images were used to develop and validate the model. Insufficient training datasets do not ensure sufficiently accurate model [38]. Thus, we generated augmented datasets by using VoxelMorph's deformable vector field (DVF) extraction method [39]. VoxelMorph is a deep-learning-based image transformation method that learns the DVF between two images such that a moving image can be changed into a fixed image. Before the DVF method was applied, various augmented images were created from the $CT^{1st}$ and $CT^{2nd}$ images utilizing conventional rigid augmentation, including image rotation, zoom-in, zoom-out, and flipping. For one image-set pair comprising $CT^{1st}$ and another augmented image, VoxelMorph was applied, and 10 DVF sets for one pair of images were extracted. Finally, the extracted DVF was applied to the $CT^{1st}$ image to create augmented images. The segmentation model was coded by adopting a U-net-based model (nnU-net) and trained by

using augmented images. Label segmentation of the $CT^{2nd}$ image was performed, and the accuracy of the model was evaluated.

## 2.1 Image acquisition and preprocessing

The study protocol was approved by the Institutional Review Board of Samsung Medical Center (IRB number 2019-09-119-002). We explained the research to adult male patients aged ≥18 years who underwent radiation treatment for prostate cancer between November 18, 2019, and September 24, 2021, and obtained signed consent forms from patients who expressed willingness to participate in additional CT scans. All patient CT data collected for research purposes were anonymized. Specifically, two sets of CT images and manually contoured structures (labels) ($CT^{1st}$ image, label; and $CT^{2nd}$ image, label) were collected from the 18 patients. CT images were acquired with two types of CT scanners, 27 Discovery™ CT590 RT scanners (GE Healthcare, Waukesha, WI, USA) and 9 LightSpeed RT16 scanners (GE Healthcare, Waukesha, WI, USA). The voxel spacing of the CT image obtained with Discovery™ CT590 RT was $0.9766 \times 0.9766 \times 2.5$ mm³ and the pixel dimensions were $512 \times 512 \times 64$. The pixel dimensions of the CT image acquired with LightSpeed RT16 were $512 \times 512 \times 64$; however, the voxel spacing was $1.2695 \times 1.2695 \times 2.5$ mm³; thus, voxel spacing was matched with the Discovery™ CT590 RT scanner through preprocessing. The VoxelMorph augmentation method learns the DFVs in the network in three dimensions (3D); hence, images with pixel dimensions of $512 \times 512 \times 64$ cause memory problems. Therefore, CT images were resized to $256 \times 256 \times 64$ pixels, and the final voxel spacing was changed to $2.5380 \times 2.5380 \times 2.5$ mm³. The bladder, rectum, and planning target volumes were determined for the segmented structures. The labels were contoured by a dosimetrist and a physician at Samsung Medical Center.

## 2.2 Augmentation

VoxelMorph is a deep-learning-based framework adopted for the deformable medical image registration of two images, wherein a moving image is deformed and registered to a fixed image through an iterative learning process by minimizing a loss function. The input image pair included a $CT^{moving}$ image and a $CT^{fixed}$ image, and each voxel of $CT^{moving}$ was deformed and converted to be similar to that in $CT^{fixed}$ via an iterative registration process in Voxel-Morph. Subsequently, multiple DVFs were created. In the learning process, DVFs were extracted by employing U-net, and a $CT^{moved}$ image was created via spatial transformation of the $CT^{moving}$ image by applying the extracted DVFs. The similarity between the $CT^{moved}$ and $CT^{fixed}$ images was scored as a loss score to transform $CT^{moving}$ to be similar to $CT^{fixed}$ through iterative training. The loss function is expressed as follows:

$$L_{main}(CT^{fixed}, CT^{moving}, \phi) = L_{MSE}(CT^{fixed}, CT^{moving} \circ \phi) + \lambda L_{smooth}(\phi), \quad (1)$$

where $\phi$ is the DVF, $CT^{moving} \circ \phi$ is the $CT^{moved}$ image transformed through spatial transform, and $L_{smooth}$ penalizes the local spatial variations in $\phi$, wherein $\lambda$ is a regularization parameter. $L_{MSE}(CT^{fix}, CT^{moving} \circ \phi)$ is the mean square voxel difference applicable when $CT^{fix}$ and $CT^{moved}$ ($= CT^{moving} \circ \phi$) have similar image intensity distributions and local contrast and is expressed as follows:

$$L_{MSE}(CT^{fix}, CT^{moving} \circ \phi) = \sum_{p \in \Omega} [CT^{fix}(p) - [CT^{moving} \circ \phi](p)]^2. \quad (2)$$

Repeated model training generated a DVF for each iteration until $CT^{moving}$ was equal to $CT^{fixed}$. The $CT^{1st}$ image with label information was utilized for $CT^{moving}$. As illustrated in Fig 1, to generate various types of augmentation data, the $CT^{2nd}$ image and 10 modified CT

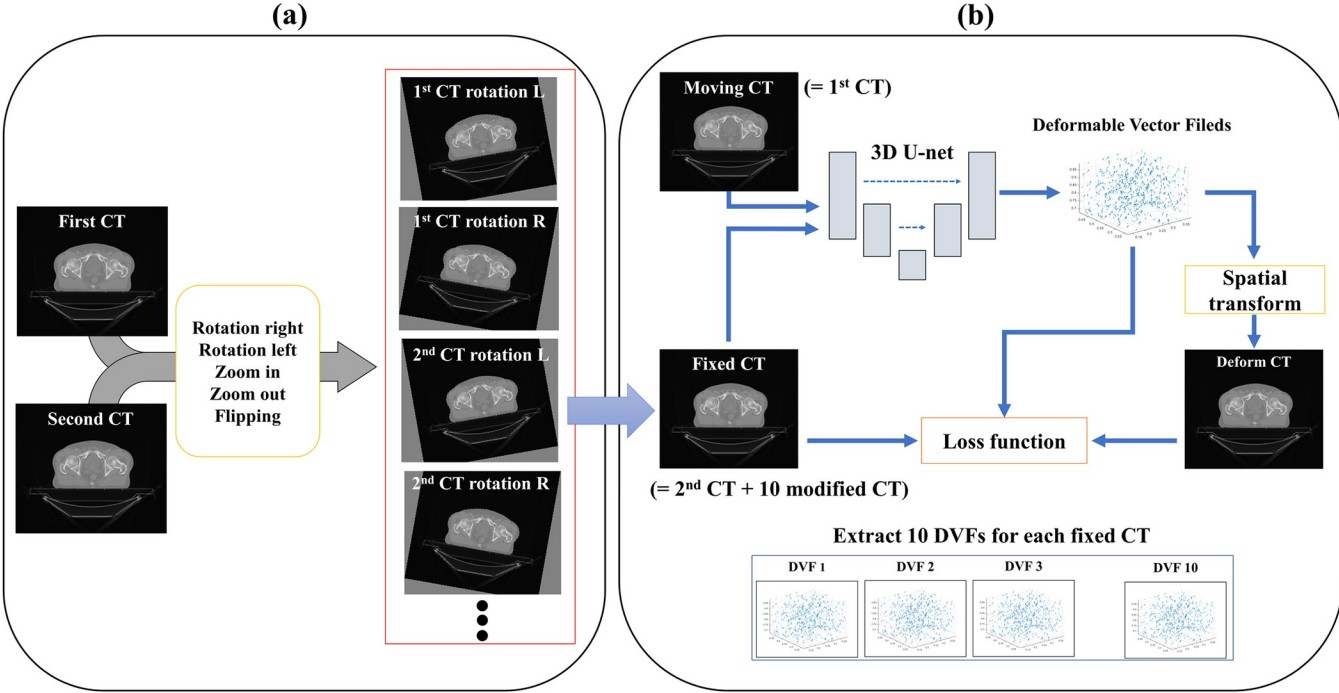

**Fig 1. Generation of DVFs utilizing VoxelMorph method.** (a) Generation of 10 modified CT images from 2 CT images of a patient to be employed as $CT^{fixed}$ images. (b) 10 DVFs extracted from each $CT^{fixed}$ via VoxelMorph method. Utilizing the 10 modified CT images and $CT^{2nd}$ image as a fixed image, 110 DVFs were extracted.

images from $CT^{1st}$ and $CT^{2nd}$ were generated by using the following parameters: right rotation 3˚, left rotation 3˚, flipping, zoom in 5%, and zoom out 5%; these were utilized as $CT^{fixed}$. When the loss exceeded 1000, the organs in the generated $CT^{moved}$ overlapped, resulting in images that did not resemble the human anatomy. Therefore, we began saving the DVF as the loss decreased to below 1000. As the iterations continued, nine more DVFs were saved such that 10 DVFs were extracted for one image pair. Each epoch consisted of 250 iterations, and the training was terminated when 10 sets of DVFs with a loss below 1000 were completely saved. Approximately 3–10 min was required to save 10 sets of DVFs. In total, 110 DVFs were generated from 11 pairs of image sets (11 $CT^{fixed}$ and one $CT^{moving}$), and 110 augmentation datasets were generated via spatial transformation of $CT^{moving}$ image and $CT^{moving}$ label with 110 DVFs (Fig 2A). Thus, 110 training sets were created utilizing $CT^{1st}$ image with labels and $CT^{2nd}$ image, which were employed as training data for the segmentation network.

## 2.3 Segmentation model

We constructed a segmentation network by adopting nnU-net [40]. nnU-net is a U-net-based segmentation model used to optimize the parameters by analyzing 23 public datasets utilized in international biomedical segmentation competitions. By employing 110 augmented data points generated through VoxelMorph, we trained the network and segmented and generated labels in $CT^{2nd}$ (Fig 2). The network utilized for training was a 3D nnU-net; 80% of the data were used for training and the remaining 20% were employed for validation. In addition, a general augmentation method, such as gamma, mirror, crop, or rotation, was randomly applied to the data for network segmentation training. The input patch size was $224 \times 224 \times 54$, and the batch size was two. Five downsampling operations were performed, and the size of

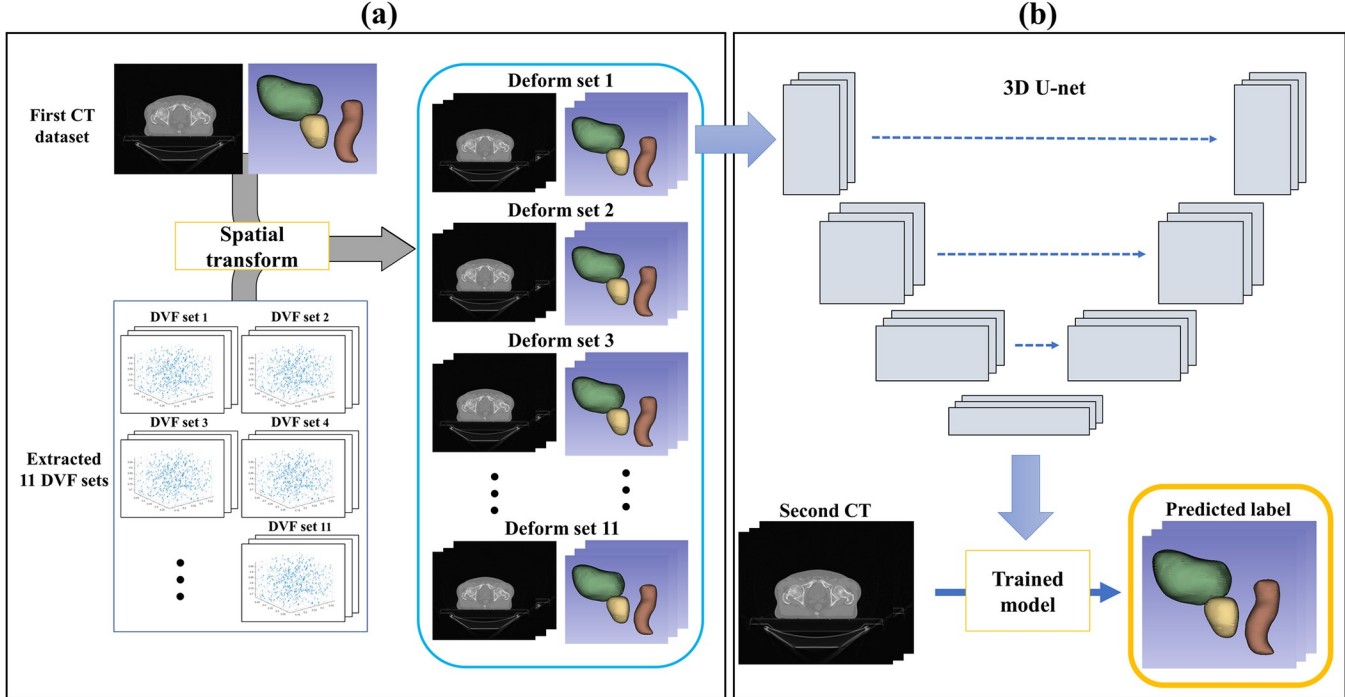

**Fig 2. CT label segmentation from deformed datasets.** (a) Deformed dataset generation through spatial transform of $CT^{1st}$ dataset with extracted DVF sets. (b) $CT^{2nd}$ label segmentation utilizing nn-Unet.

the function map was $4 \times 4 \times 3$ at the bottleneck. The optimizer employed was a stochastic gradient descent, and the loss was computed using the Dice and cross-entropy scores. The training was run for 50 epochs, and each epoch was equal to 250 iterations. The learning rate was constantly reduced by using a polynomial. The training was performed by applying Voxel-Morph in Pytorch 1.5.1 and nnU-net in TensorFlow 2.0, employing an NVIDIA GeForce 2080Ti graphics processing unit. For 50 epochs, we saved the model at each epoch. Among the 50 models, the model with the lowest Dice and cross-entropy loss on the validation data was selected as the final model for the patient; thus, 18 segmentation models were produced.

## 2.4 Evaluation

We conducted segmentation using a limited individual dataset and VoxelMorph augmentation, which referred to an individual model. To assess the impact of the number of datasets and the use of VoxelMorph on the model performance, we compared the segmentation performance using individual datasets against the total dataset (data from 18 people). The model trained with all datasets is referred to as the total model. Additionally, we evaluated the effect of incorporating voxel-morphs on model performance. We also tested the dependency of the model performance on the network type by comparing the Basic U-net and nnU-net. For this purpose, a U-net-based model was developed and trained using a process identical to that adopted for the nnU-net-based model. The predicted segmentation was quantitatively evaluated utilizing the DSC, Hausdorff distance (HD), and mean surface distance (MSD). The DSC evaluates the overlap between true and predicted volumes.

$$DSC = \frac{2|True \cap Pred|}{|True| + |Pred|},$$ 

(3)

where True and Pred denote the actual and predicted label volumes, respectively. Meanwhile, the HD is the largest distance from a point on the true label to the nearest point on the predicted label and is expressed as follows:

$$HD = Max(h_{HD}(True, Pred), h_{HD}(Pred, True)); \qquad (4)$$

$$h_{HD}(True, Pred) = \max_{t \in True} \min_{p \in Pred} \|t - p\|, \qquad (5)$$

where $h_{HD}(True, Pred)$ is the Euclidean distance between two points: one in the true label volume and the other in the predicted label volume. The MSD measures the average surface changes between the predicted and true label volumes and is expressed as follows:

$$MSD = \frac{1}{2}(h_{MSD}(True, Pred) + h_{MSD}(Pred, True)); \qquad (6)$$

$$h_{MSD}(True, Pred) = \frac{1}{True} \sum_{t \in True} \min_{p \in Pred} \|t - p\|. \qquad (7)$$

## 3. Results

Fig 3 presents examples of the predicted $CT^{2nd}$ labels using individual models and the actual $CT^{2nd}$ labels of the prostate, rectum, and bladder, presented as the 3D surface of the organ utilizing a 3D slice viewer [41]. Six cases show the highest or lowest DSC for each organ. For a

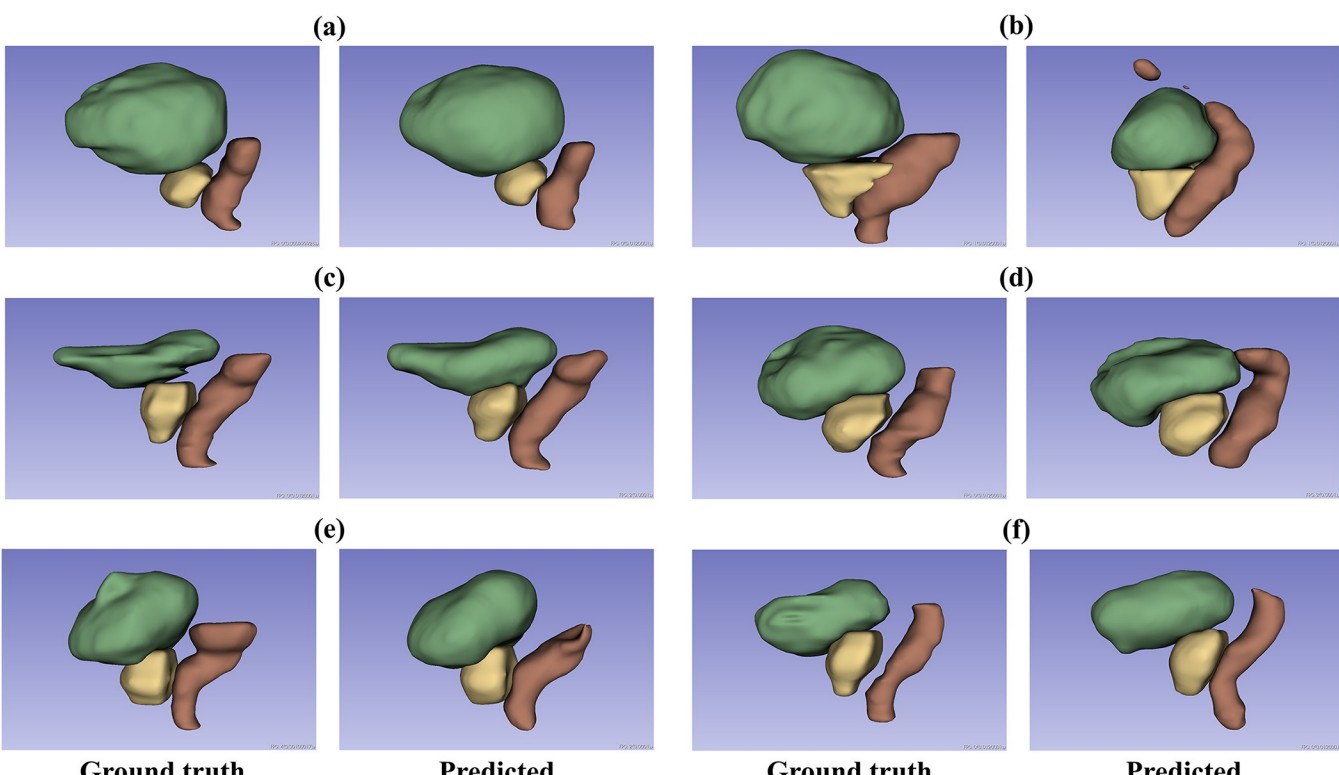

**Fig 3. Segmentation comparison using 3D volumes.** (A) Case 4 with the highest bladder (green colored organ) DSC of 0.969. (B) Case 14 with the lowest bladder DSC of 0.881. (C) Case 11 with the highest rectum (brown colored organ) DSC of 0.904. (D) Case 10 with the lowest rectum DSC of 0.753. (E) Case 3 with the highest prostate (yellow colored organ) DSC of 0.933. (F) Case 13 with the lowest prostate DSC of 0.673.

## Case 01                                    Case 07

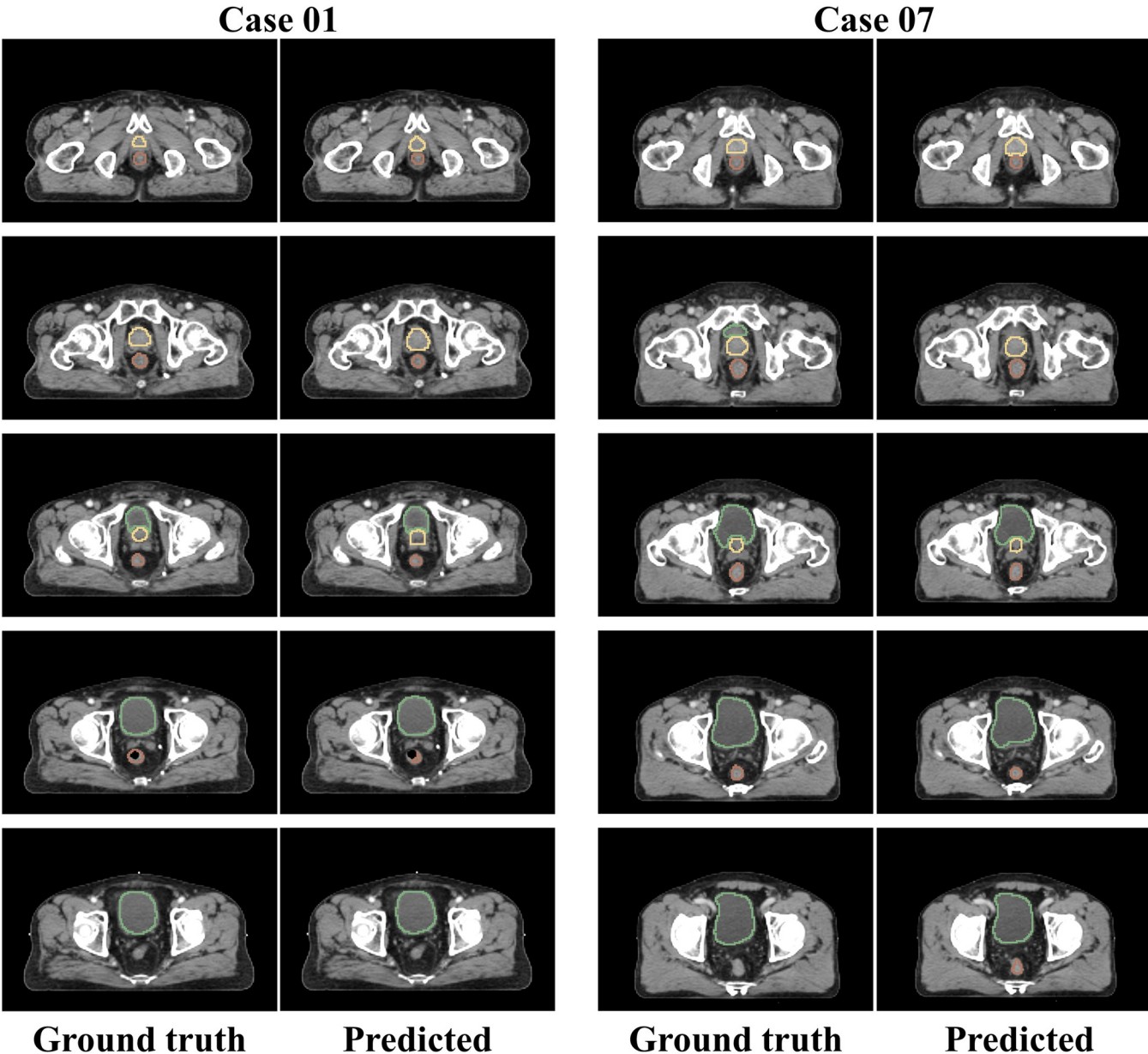

**Ground truth          Predicted          Ground truth          Predicted**

**Fig 4. Comparison of segmentation by overlapping CT and segmentation.** Case 01 contains air in the rectum; Case 07 contains no air in the rectum.

more intuitive comparison, the predicted and actual $CT^{2nd}$ labels of Cases 01 and 07 were overlaid on each $CT^{2nd}$ axial image, as illustrated in Fig 4.

Table 1 presents a comparison between the performances of VoxelMorph for different types of datasets. In the nnU-net, the highest values of 0.957 and 0.874 were presented for the bladder and rectum, respectively, in the total dataset with VoxelMorph. Whereas, for the prostate, the highest value of 0.847 was realized in the individual dataset without VoxelMorph. The performance of Basic U-net was lower than that of nnU-net; however, the best-performing combination for each organ remained the same. In addition, the DSC values for the bladder, rectum, and prostate were computed on $CT^{2nd}$ by adopting the labels generated in the $CT^{1st}$ ($CT^{1st}$ to $CT^{2nd}$ segmentation). The rectum exhibited the lowest value (0.624) of the three labels.

**Table 1. Comparison of DSC values with different networks, dataset numbers, VoxelMorph.**

| Network | Dataset | VoxelMorph | Data number | Bladder | Rectum | Prostate |
|---|---|---|---|---|---|---|
| | $CT^{1st}$ label to $CT^{2nd}$ segmentation | | | 0.715 ± 0.11 | 0.624 ± 0.15 | 0.740 ± 0.17 |
| nnU-net | Individual | Yes | 2* | 0.944 ± 0.03 | 0.832 ± 0.04 | 0.840 ± 0.07 |
| | Individual | No | 1 | 0.928 ± 0.05 | 0.831 ± 0.05 | **0.847 ± 0.08** |
| | Total | Yes | 36* | **0.957 ± 0.02** | **0.874 ± 0.02** | 0.838 ± 0.07 |
| | Total | No | 18 | 0.955 ± 0.02 | 0.856 ± 0.04 | 0.835 ± 0.07 |
| Basic U-net | Individual | Yes | 2* | 0.904 ± 0.04 | 0.807 ± 0.04 | 0.806 ± 0.08 |
| | Individual | No | 1 | 0.912 ± 0.04 | 0.807 ± 0.04 | **0.815 ± 0.08** |
| | Total | Yes | 36* | **0.925 ± 0.03** | **0.825 ± 0.03** | 0.814 ± 0.08 |
| | Total | No | 18 | 0.919 ± 0.03 | 0.820 ± 0.03 | 0.811 ± 0.07 |

'Individual' refers to a model utilizing an individual dataset, with or without the use of VoxelMorph, and 'total' pertains to a model employing the entire datasets, with or without the use of VoxelMorph. DSC, disc similarity coefficient, * one labeled CT image set and one unlabeled CT image set. Bold text indicates the highest DSC.

The DSC, HD, and MSD scores for the three organs automatically segmented using the nnU-net network with VoxelMorph were computed for 18 patients in both the individual and total models. The performances of the individual and total models are summarized in Tables 2 and 3, respectively. In the DSC evaluations of the two models, the bladder had average DSC values of 0.944 and 0.957, with the highest DSC values among the segmented organs. In terms

**Table 2. DSC, HD, and MSD performance evaluation of individual models in each patient.**

| Matrix | DSC | | | HD [mm] | | | MSD [mm] | | |
|---|---|---|---|---|---|---|---|---|---|
| Case | Bladder | Rectum | Prostate | Bladder | Rectum | Prostate | Bladder | Rectum | Prostate |
| 1 | 0.958 | 0.811 | 0.794 | 3.906 | 8.951 | 6.478 | 0.420 | 0.953 | 1.547 |
| 2 | 0.964 | 0.841 | 0.917 | 14.616 | 5.860 | 2.762 | 0.530 | 1.014 | 0.564 |
| 3 | 0.947 | 0.759 | 0.933 | 9.959 | 13.102 | 2.762 | 0.670 | 1.337 | 0.467 |
| 4 | 0.969 | 0.876 | 0.903 | 4.367 | 4.367 | 2.762 | 0.439 | 0.665 | 0.551 |
| 5 | 0.956 | 0.838 | 0.752 | 5.524 | 6.177 | 8.287 | 0.628 | 0.697 | 2.379 |
| 6 | 0.946 | 0.825 | 0.908 | 5.860 | 5.860 | 5.860 | 0.486 | 0.781 | 1.002 |
| 7 | 0.961 | 0.831 | 0.897 | 4.784 | 15.747 | 2.762 | 0.469 | 1.242 | 0.681 |
| 8 | 0.961 | 0.872 | 0.763 | 5.524 | 4.367 | 7.042 | 0.517 | 0.549 | 2.058 |
| 9 | 0.912 | 0.856 | 0.708 | 5.524 | 5.524 | 5.860 | 0.500 | 0.761 | 1.881 |
| 10 | 0.946 | 0.753 | 0.906 | 5.860 | 22.778 | 4.367 | 0.633 | 2.238 | 0.703 |
| 11 | 0.902 | 0.904 | 0.812 | 22.610 | 5.860 | 5.860 | 0.844 | 0.547 | 1.379 |
| 12 | 0.950 | 0.789 | 0.846 | 4.367 | 9.766 | 4.367 | 0.618 | 1.207 | 1.029 |
| 13 | 0.915 | 0.795 | 0.673 | 8.514 | 9.766 | 8.053 | 0.830 | 0.994 | 2.634 |
| 14 | 0.881 | 0.853 | 0.839 | 15.380 | 23.520 | 6.177 | 2.071 | 1.814 | 0.930 |
| 15 | 0.968 | 0.896 | 0.890 | 2.762 | 7.308 | 5.860 | 0.298 | 0.660 | 0.809 |
| 16 | 0.942 | 0.864 | 0.813 | 5.860 | 5.860 | 5.860 | 0.588 | 0.652 | 1.431 |
| 17 | 0.949 | 0.798 | 0.894 | 19.434 | 7.308 | 4.367 | 0.806 | 1.229 | 0.769 |
| 18 | 0.967 | 0.815 | 0.870 | 4.367 | 8.951 | 3.906 | 0.493 | 0.933 | 0.790 |
| Min | 0.881 | 0.753 | 0.673 | 2.762 | 4.367 | 2.762 | 0.298 | 0.547 | 0.467 |
| Max | 0.969 | 0.904 | 0.933 | 22.610 | 23.520 | 8.287 | 2.071 | 2.238 | 2.634 |
| Average | 0.944 | 0.832 | 0.840 | 8.290 | 9.504 | 5.188 | 0.658 | 1.015 | 1.200 |
| Standard deviation | 0.024 | 0.042 | 0.074 | 5.634 | 5.616 | 1.722 | 0.371 | 0.436 | 0.640 |

DSC, Dice similarity coefficient; HD, Hausdorff distance; MSD, mean surface distance.

**Table 3. DSC, HD, and MSD performance evaluation of total model in each patient.**

| Matrix | DSC | | | HD [mm] | | | MSD [mm] | | |
|---|---|---|---|---|---|---|---|---|---|
| Case | Bladder | Rectum | Prostate | Bladder | Rectum | Prostate | Bladder | Rectum | Prostate |
| 1 | 0.961 | 0.906 | 0.776 | 5.523 | 3.906 | 7.042 | 0.394 | 0.467 | 1.675 |
| 2 | 0.968 | 0.852 | 0.906 | 13.671 | 5.859 | 3.906 | 0.460 | 0.914 | 0.676 |
| 3 | 0.954 | 0.886 | 0.902 | 9.568 | 3.906 | 2.762 | 0.569 | 0.639 | 0.662 |
| 4 | 0.972 | 0.906 | 0.898 | 3.906 | 4.367 | 3.383 | 0.398 | 0.484 | 0.595 |
| 5 | 0.967 | 0.856 | 0.773 | 4.367 | 4.367 | 7.042 | 0.482 | 0.623 | 2.061 |
| 6 | 0.947 | 0.855 | 0.915 | 4.367 | 4.367 | 4.367 | 0.512 | 0.680 | 0.901 |
| 7 | 0.969 | 0.861 | 0.848 | 3.906 | 5.859 | 4.367 | 0.380 | 0.634 | 1.052 |
| 8 | 0.961 | 0.897 | 0.747 | 4.784 | 4.367 | 6.176 | 0.534 | 0.457 | 2.182 |
| 9 | 0.910 | 0.895 | 0.709 | 5.524 | 4.367 | 5.859 | 0.500 | 0.515 | 1.874 |
| 10 | 0.968 | 0.849 | 0.931 | 3.383 | 17.685 | 2.762 | 0.365 | 1.189 | 0.500 |
| 11 | 0.918 | 0.910 | 0.815 | 4.367 | 3.906 | 5.859 | 0.671 | 0.473 | 1.424 |
| 12 | 0.958 | 0.834 | 0.873 | 2.762 | 4.784 | 4.367 | 0.506 | 0.915 | 0.865 |
| 13 | 0.934 | 0.837 | 0.685 | 5.524 | 7.042 | 8.286 | 0.633 | 0.750 | 2.524 |
| 14 | 0.976 | 0.881 | 0.850 | 2.762 | 9.765 | 7.042 | 0.354 | 0.967 | 0.893 |
| 15 | 0.966 | 0.883 | 0.880 | 2.762 | 7.812 | 7.042 | 0.317 | 0.743 | 0.935 |
| 16 | 0.952 | 0.882 | 0.822 | 5.524 | 8.286 | 5.524 | 0.512 | 0.594 | 1.355 |
| 17 | 0.968 | 0.846 | 0.908 | 3.906 | 5.859 | 2.762 | 0.395 | 0.955 | 0.669 |
| 18 | 0.972 | 0.891 | 0.853 | 2.758 | 3.906 | 4.367 | 0.418 | 0.547 | 0.934 |
| Min | 0.910 | 0.834 | 0.685 | 2.758 | 3.906 | 2.762 | 0.317 | 0.457 | 0.500 |
| Max | 0.976 | 0.910 | 0.931 | 13.671 | 17.685 | 8.286 | 0.671 | 1.189 | 2.524 |
| Average | 0.957 | 0.874 | 0.838 | 4.965 | 6.134 | 5.162 | 0.467 | 0.697 | 1.210 |
| Standard deviation | 0.018 | 0.024 | 0.072 | 2.631 | 3.269 | 1.676 | 0.095 | 0.206 | 0.595 |

DSC, Dice similarity coefficient; HD, Hausdorff distance; MSD, mean surface distance.

of the average HD, the individual model had the lowest values in the order of prostate, bladder, and rectum, while the total model had the lowest values in the order of bladder, prostate, and rectum. The average MSDs of 0.658 and 0.467 mm were observed in the bladder (lowest), followed by 1.015 and 0.697 mm in the rectum, and 1.200 and 1.210 mm in the prostate.

Although a direct comparison is not possible, Table 4 summarizes recently published studies that evaluated model performance using DSC values to compare the quantities of data utilized. The dataset number refers to the number of CT images with labels. This study employed a single dataset and an unlabeled CT image set to train the segmentation model. Among the studies evaluated, Sultana et al. [42] achieved the highest DSC value of 0.90 ± 0.05 for the prostate. Kiljunen et al. [43] employed the largest number of datasets, specifically 876, for training the segmentation model; whereas, Kawula et al. [44] utilized the smallest number of datasets, 47, excluding our study. The performance of the model developed using one dataset was as follows: eighth among the 12 models for the prostate, eighth among the 10 models for the rectum, and fourth among the 10 models for the bladder. The performance of the model developed by utilizing the total dataset was the same for the prostate and bladder as that by utilizing the individual datasets; however, the ranking of the rectum increased from eighth to fourth among the 10 models.

To utilize the image generated by using VoxelMorph, whether the image resembles an actual human body should be determined. Fig 5 illustrates 12 of the 110 augmented images generated by using the VoxelMorph model in Case 04 from the same slice number. These 12 images depict various shapes of the human body.

**Table 4. Comparison of DSC values with state-of-the-art.**

| Reference | Year | Network | Dataset number | Bladder | Rectum | Prostate |
|---|---|---|---|---|---|---|
| [45] | 2019 | 3D U-net | 774 | | | 0.88 ± 0.03 |
| [43] | 2020 | Transform 3D U-net | 738–876 | 0.93 | 0.84 | 0.82 |
| [42] | 2020 | 3D U-net + GAN | 100 | 0.96 ± 0.06 | 0.91 ± 0.09 | 0.90 ± 0.05 |
| [46] | 2020 | Transform 3D U-net | 259 | 0.91 | 0.83 | 0.83 |
| [47] | 2020 | 2D U-net | 500 | 0.94 ± 0.04 | 0.85 ± 0.07 | 0.85 ± 0.05 |
| [48] | 2021 | 2D U-net + GAN | 100 | 0.97 ± 0.07 | 0.86 ± 0.12 | 0.88 ± 0.11 |
| [49] | 2021 | Transform 2D U-net | 149 | 0.88 | 0.80 | 0.76 |
| [50] | 2021 | Transform 2D U-net | 313 | 0.94 ± 0.01 | 0.92 ± 0.01 | 0.90 ± 0.01 |
| [51] | 2021 | Transform 2D U-net | 241 | | | 0.88 ± 0.03 |
| [44] | 2022 | 3D U-net | 47 | 0.97 ± 0.01 | 0.89 ± 0.04 | 0.87 ± 0.03 |
| [52] | 2022 | 3D U-net | 84 | 0.93 ± 0.04 | 0.85 ± 0.05 | 0.83 ± 0.05 |
| Average | | | 306 | 0.94 ± 0.03 | 0.86 ± 0.04 | 0.85 ± 0.04 |
| Our | | Individual model | 2* | 0.94 ± 0.02 | 0.83 ± 0.04 | 0.84 ± 0.07 |
| | | Total model | 36* | 0.96 ± 0.02 | 0.87 ± 0.02 | 0.84 ± 0.07 |

* One labeled CT image set and one unlabeled CT image set.

## 4. Discussion

ART can improve therapeutic efficacy because a treatment plan can be adapted by continuously updating the shapes and positions of the patient's OAR and target in the treatment plan. Accordingly, we created and evaluated a patient-specific automatic segmentation model utilizing labeled and additional unlabeled CT images. Performing accurate automatic segmentation is technically challenging because the volumes of the prostate and rectum are small, and the Hounsfield unit value on CT images is similar to that of soft tissues. Nevertheless, employing one individual dataset, the proposed model achieved good performance; the DSC values were more than 0.80 for the prostate and rectum and 0.94 for the bladder. These results demonstrate the feasibility of using a patient-specific segmentation model with an individual patient dataset.

We compared the performance of the employed model, nnU-net, against a Basic U-net with varying numbers of datasets, with and without a VoxelMorph (Table 1). For the individual datasets, the nnU-net network using VoxelMorph performed better than that without VoxelMorph. Conversely, the Basic U-net network model performed better without a VoxelMorph. However, for the total dataset, VoxelMorph performed the best for both network models. VoxelMorph augmentation increases the segmentation performance with a more advanced network model, particularly when models are trained with larger datasets (total model).

As shown in Table 1, the segmented labels of $CT^{2nd}$ adopting the $CT^{1st}$ label exhibited the lowest DSC, in the order of the rectum, bladder, and prostate. Compared with the segmentation results utilizing deep learning, the DSC values for the rectum and bladder increased by more than 0.20. This suggests that the shape changes of the organs according to the progression of radiation treatment are minimal in the prostate and notable in the bladder and rectum.

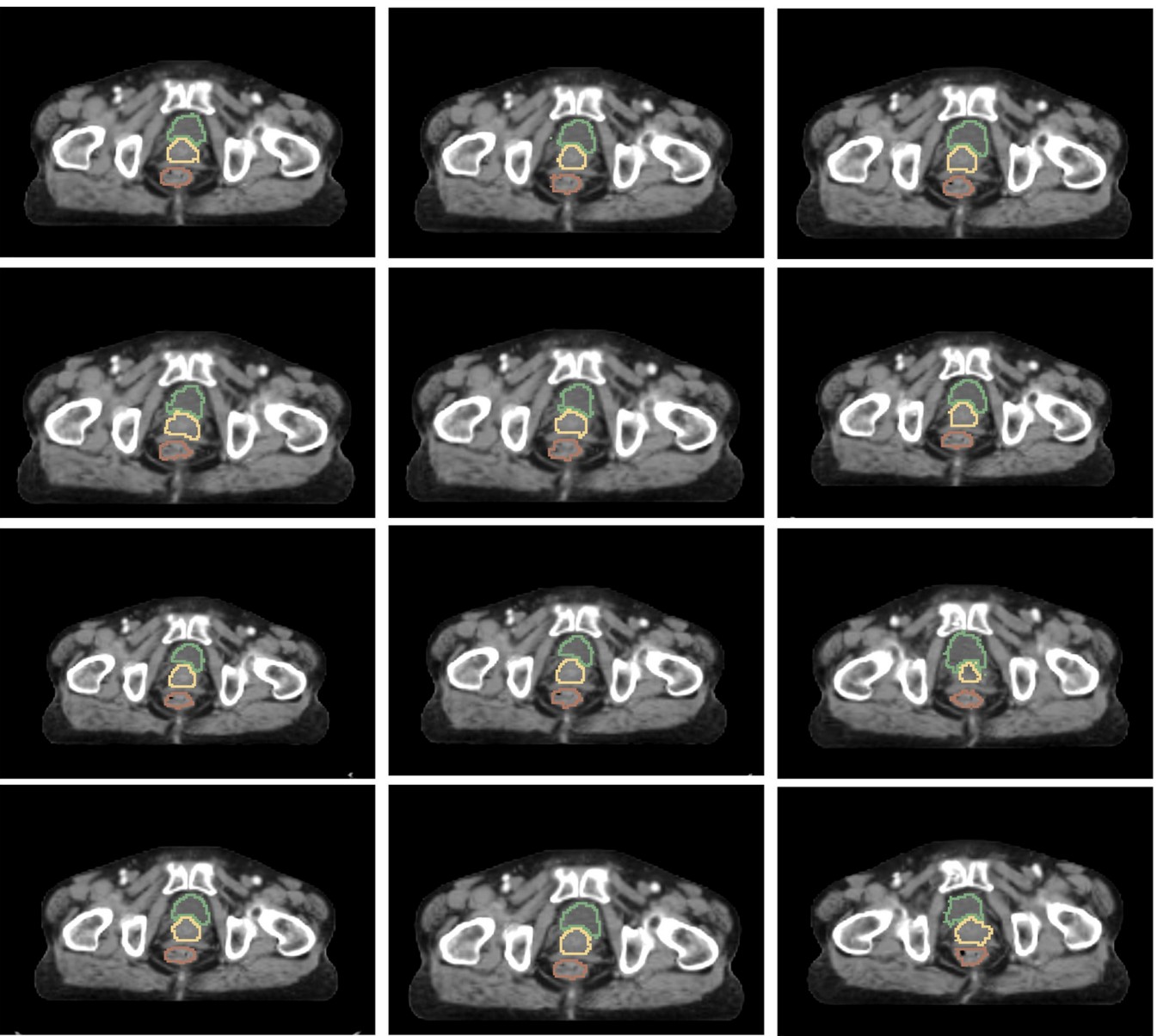

**Fig 5. Augmented CT images and labels generated in Case 04 utilizing VoxelMorph.** To verify whether the generated image exhibits a similar structure to the actual image, an examination of the image generated using VoxelMorph is presented for patient case 04.

As depicted in Fig 3, the shape of the rectum varied in each patient. VoxelMorph augmentation in the total model demonstrated superior performance compared with the individual model, attributed to its capacity to learn from diverse rectal shapes. This leads to enhanced rectal segmentation performance.

The average DSC values obtained in this study were compared to those reported in other studies (Table 4). The average DSC values for the bladder and prostate in both the developed individual and total models closely match those reported in previous studies. However, for the rectum, the individual model exhibited a lower performance, while the total model exhibited a high performance. For the bladder, our total model exhibited a DSC value of 0.02 higher than the average reported in the referenced studies, where a DSC value of 0.90 or more was

observed. Conversely, the individual model's rectum had a DSC value 0.03 lower than the average reported, falling below the values in all referenced studies, except for a DSC value of 0.8 in the study conducted by Xu et al. [49]. However, the DSC value was 0.01 higher than the average reported. Both the individual and total models for the prostate showed DSC values lower than the average reported in the referenced studies. The DSC values observed in Case 13 failed to accurately distinguish the boundary between the prostate and rectum (Fig 3F).

The DSC value reported by Kiljunen et al. [43], who employed 800 datasets, was similar to that obtained for the individual models, and the DSC value reported by Elmahdy et al. [46], who employed 259 datasets, was higher than that obtained for all organs in the individual models. The performance of the aforementioned models does not depend linearly on the amount of data used. Considering that the studies cited in Table 4 did not employ the same dataset, we found that the data quantity, quality, network structures, and other parameters could affect the accuracy of the model. Nevertheless, quantitative measures of the performance of the developed individual models proved the feasibility of automatic segmentation by adopting only one patient dataset.

The HD is the maximum distance from a point on the true label to the nearest point on the predicted label. The HD values of the segmented organs indicate the presence of outliers. In the individual model, the average HD was < 10 mm for all organs, while the maximum HDs in the bladder and rectum exceeded 20 mm, with the standard deviation of 5 mm. Conversely, in the total model, the maximum HDs of the bladder and rectum were within 20 mm, with the standard deviation of 3 mm. The bladder and rectum exhibited fewer outliers as the number of datasets increased. Compared with the study conducted by Liu et al. [45], where the reported average and maximum HDs for the prostate were 7.0 and 18.4 mm, respectively, our individual and total models achieved average and maximum HDs for the prostate of 5.19 and 8.29 mm, and 5.16 and 8.29 mm, respectively. This suggests that the performance of the proposed model is comparable to that of Liu's model using 774 datasets, indicating a minimal impact of the dataset number on prostate segmentation. A small HD indicates small error of the outliers, thereby demonstrating that our segmentation method produced predictions that aligned closely with the actual organ.

The MSD represents the mean distance error between the actual and predicted labels. The average MSDs of the bladder and rectum in both datasets were comparable: the rectum exhibited a lower MSD in the total model, with the average difference of 0.318 mm. An increase in performance was realized regarding the rectum using an augmented and increased dataset; whereas, for the bladder and prostate, similar performances were exhibited as the total model, even with one dataset. Moreover, in more than 50% of the increases in the individual model, specifically 9 of 18 cases in the prostate and 11 of 18 cases in the rectum, the MSD was less than 1 mm. Notably, the recommended treatment accuracy of radiation therapy by Task Group 142 of the American Association of Physicists in Medicine is 1 mm [53]. Although the criteria for treatment accuracy differ from those for MSD, more than 50% of the individual models meet the treatment accuracy criteria. Segmentation using individual models demonstrates the potential for actual treatment use.

Our study has several limitations. We utilized simulated CT data and an additional unlabeled CT image taken during treatment. Additional CT scans are difficult to perform during the treatment for patients undergoing short-term treatments such as stereotactic body radiotherapy. If the improvement in the cone-beam CT images taken for setup verification is equivalent to the simulation CT images, additional CT scans can be eliminated. In Case 01, as illustrated in Fig 4, the model failed to predict the air area of the rectum. The training $CT^{1st}$ for the rectum lacked an air area and air label, resulting in the model's inability to predict the air area in the $CT^{2nd}$. Therefore, predicting the air area on $CT^{2nd}$ is feasible if at least one dataset

contains air in the rectal label. Nevertheless, our model demonstrated a performance comparable to that of the models proposed in studies utilizing many datasets. Further studies are necessary to enhance accuracy by refining the segmentation model and diversifying the DVF-based augmentation method. The prostate, rectum, and bladder considered in this study showed fewer morphological changes over time than the other organs. Our augmentation method learns these shape changes to generate images; therefore, minor morphological changes may make producing diverse images difficult. Additionally, we plan to recruit patients and focus on other organs to compare and validate our augmentation-based segmentation.

## 5. Conclusion

We developed a personalized automatic segmentation method utilizing individual patient datasets and image augmentation techniques to customize radiation therapy based on variations in the target and critical organs of patients with prostate cancer. In contrast to previous studies in which large datasets were utilized, in this study, we demonstrated the feasibility of automatic segmentation by adopting only two datasets per patient. Thus, this model is a promising tool for ART in patients with prostate cancer.

## Author Contributions

**Conceptualization:** Wonjoong Cheon, Won Park, Youngyih Han.

**Data curation:** Sangwoon Jeong, Wonjoong Cheon, Sungjin Kim, Won Park.

**Formal analysis:** Sangwoon Jeong.

**Methodology:** Wonjoong Cheon, Sungjin Kim, Youngyih Han.

**Project administration:** Youngyih Han.

**Resources:** Youngyih Han.

**Software:** Sangwoon Jeong.

**Supervision:** Youngyih Han.

**Validation:** Youngyih Han.

**Writing – original draft:** Sangwoon Jeong.

**Writing – review & editing:** Youngyih Han.

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
