## [Decision Letter · Decision Letter 0]

13 May 2024

PONE-D-23-40530Deep-learning-based segmentation using individual patient data of prostate cancer radiation therapy.PLOS ONE

Dear Dr. Han,

Thank you for submitting your manuscript to PLOS ONE. After careful consideration, we feel that it has merit but does not fully meet PLOS ONE’s publication criteria as it currently stands. Therefore, we invite you to submit a revised version of the manuscript that addresses the points raised during the review process.

We look forward to receiving your revised manuscript.

Kind regards,

Khan Bahadar Khan, Ph.D

Academic Editor

PLOS ONE

Journal Requirements:

This research was supported by the National Research Foundation of Korea grants funded by the Korean government (MSIT) (2022R1A2C2004694 and 2021M28A1048108).

@ Initials of the authors who received each award: Y.Han

@ Grant numbers: 2022R1A2C2004694 and 2021M28A1048108

@The full name of each funder:  National Research Foundation of Korea grants funded by the Korean government (MSIT)

@URL of each funder website: https://www.nrf.re.kr/eng/index

@Did the sponsors or funders play any role in the study design, data collection and analysis, decision to publish, or preparation of the manuscript?: No

5. Thank you for uploading your study's underlying data set. Unfortunately, the repository you have noted in your Data Availability statement does not qualify as an acceptable data repository according to PLOS's standards.

Reviewers' comments:

Reviewer's Responses to Questions

**Comments to the Author**

1. Is the manuscript technically sound, and do the data support the conclusions?

Reviewer #1: Yes

Reviewer #2: Yes

2. Has the statistical analysis been performed appropriately and rigorously? 

Reviewer #1: Yes

Reviewer #2: Yes

3. Have the authors made all data underlying the findings in their manuscript fully available?

Reviewer #1: No

Reviewer #2: Yes

4. Is the manuscript presented in an intelligible fashion and written in standard English?

Reviewer #1: No

Reviewer #2: Yes

5. Review Comments to the Author

**Reviewer #1:** 1. The methodology is well-described, but considering potential limitations like variability in patient anatomy could strengthen the paper's depth.

2. The discussion effectively underscores the significance of the segmentation method for prostate cancer radiation therapy. Briefly discussing future directions like incorporating additional clinical variables or validating across diverse patient populations would enrich the conclusion.

3. Highlighting the feasibility of automatic segmentation with two datasets per patient is commendable. However, discussing encountered challenges and strategies to address them would provide a more balanced perspective, enhancing the paper's contribution to adaptive radiotherapy.

4. While the methodology is comprehensive, a brief discussion on the evaluation of final segmentation models would enhance clarity. Detailing the criteria used for selecting the best-performing model among 50 epochs would ensure transparency and reproducibility.

5. The article needs major improvements in English language and there are many grammars and sentence error in this paper.

**Reviewer #2:** This paper describes a technique of creating individualized patient neural networks for organ segmentation in patients receiving radiation therapy for prostate cancer. It is well conceived and executed and well written. Its main strength is the utilization of data augmentation to demonstrate good segmentation performance with very limited training data. The technical aspects are clearly described and many examples are clearly presented. Its main limitation is the limited number of patients (18) included.

A few specific comments/questions:

1. Were all patients who presented for radiation treatment at Samsung Medical Center between November 18, 2019 and September 24, 2021 included in the sample? If there were any patients excluded please explain why/exclusion criteria. Potential bias in such a small sample may limit generalizability of findings.

2. In methods you state "if the loss decreased below 1000 and the generated image was similar to that of a human body, the DVFs were saved." Please clarify how "similar to that of a human body" was defined - was this by manual visual inspection? If so, what was the time required to do this?

3. The individual models were compared with a total model containing data for just 18 patients. Is this a big enough "total model" or should further research be done to validate the individual model performance against a larger total model?

4. In your abstract conclusion you state "The proposed method can be clinically adopted for automatic prostate segmentation for adaptive radiation therapy." Do you believe this model is ready for clinical use? How would you propose it be adopted - to generate a "first draft" radiation plan which would be reviewed by a physician prior to radiation being given to the patient?

5. Please proofread for minor errors, e.g. "Abstrtact" and periods inserted in the middle of sentences after references.

6. PLOS authors have the option to publish the peer review history of their article (what does this mean?). If published, this will include your full peer review and any attached files.

Reviewer #1: No

Reviewer #2: No

---

## [Author Response · Author response to Decision Letter 0]

7 Jun 2024

Journal Requirements:

: It was modified to PLOS ONE's style through English proofreading.

: We're constantly revising code sharing (https://github.com/SangWoonJeong/DVF-based-segmentation) to validate guidelines and promote reproducibility.

The grant numbers were corrected.

This research was supported by the National Research Foundation of Korea grants funded by the Korean government (MSIT) (2022R1A2C2004694 and 2021M28A1048108).

@ Initials of the authors who received each award: Y.Han

@ Grant numbers: 2022R1A2C2004694 and 2021M2E8A1048108

@The full name of each funder: National Research Foundation of Korea grants funded by the Korean government (MSIT)

@URL of each funder website: https://www.nrf.re.kr/eng/index

@Did the sponsors or funders play any role in the study design, data collection and analysis, decision to publish, or preparation of the manuscript?: No

Thank you. The stated funding information is correct. I will include that information in the cover letter.

5. Thank you for uploading your study's underlying data set. Unfortunately, the repository you have noted in your Data Availability statement does not qualify as an acceptable data repository according to PLOS's standards.

At this time, please upload the minimal data set necessary to replicate your study's findings to a stable, public repository (such as figshare or Dryad) and provide us with the relevant URLs, DOIs, or accession numbers that may be used to access these data. For a list of recommended repositories and additional information on PLOS standards for data deposition, please see https://journals.plos.org/plosone/s/recommended-repositories

: We shared the code used in the study (at https://github.com/SangWoonJeong/DVF-based-segmentation), but not the data. Data sharing must be approved through the hospital's data review committee. The data sharing request was applied, and the review is under process. Once data sharing is approved, we will share it to the recommended repository.

: To share the image data used in this research, data open must be approved after evaluating images by the hospital's data review committee. The evaluation process takes from one to three months. Therefore, patient image data sharing will be possible after permission.

7. Please review your reference list to ensure that it is complete and correct. If you have cited papers that have been retracted, please include the rationale for doing so in the manuscript text or remove these references and replace them with relevant current references. Any changes to the reference list should be mentioned in the rebuttal letter that accompanies your revised manuscript. If you need to cite a retracted article, indicate the article’s retracted status in the References list and also include a citation and full reference for the retraction notice.

Yes, we have checked for any missing reference numbers and ensured that the content aligns well with the main text. .

Reviewer #1: 

1. The methodology is well-described, but considering potential limitations like variability in patient anatomy could strengthen the paper's depth.

Response 

Thank you for the valuable feedback. Our study has potential limitations owing to the minimal anatomical changes in the organs used over the course of treatment. Therefore, further evaluation of our research methods by using other organs is necessary.

Modified [Line 357 - 361].

The prostate, rectum, and bladder considered in this study showed fewer morphological changes over time than the other organs. Our augmentation method learns these shape changes to generate images; therefore, minor morphological changes may make producing diverse images difficult. Additionally, we plan to recruit patients and focus on other organs to compare and validate our augmentation-based segmentation. 

2. The discussion effectively underscores the significance of the segmentation method for prostate cancer radiation therapy. Briefly discussing future directions like incorporating additional clinical variables or validating across diverse patient populations would enrich the conclusion.

Response 

Thank you for your insightful comment. As mentioned in the first response, we plan to recruit additional patients with different types of cancer and conduct evaluations using radiation therapy.

Modified [Line 357 - 361].

The prostate, rectum, and bladder considered in this study showed fewer morphological changes over time than the other organs. Our augmentation method learns these shape changes to generate images; therefore, minor morphological changes may make producing diverse images difficult. Additionally, we plan to recruit patients and focus on other organs to compare and validate our augmentation-based segmentation. 

3. Highlighting the feasibility of automatic segmentation with two datasets per patient is commendable. However, discussing encountered challenges and strategies to address them would provide a more balanced perspective, enhancing the paper's contribution to adaptive radiotherapy.

Response 

Thank you for your valuable comments. In this study, obtaining two sets of planning-quality CTs from patients was the most difficult. As CBCT is performed daily, it can be used instead of 2nd set of planning-quality CT. Although the image quality of CBCT is poor, recent AI models have been developed to enhance the CBCT image quality to the level of planned CTs. We believe that daily ART will be possible if AI-based CBCT image quality upgrades are conducted in conjunction with our research. This is a limitation of the present study.

Modified [Line 348-351]

Our study has several limitations. We utilized simulated CT data and an additional unlabeled CT image taken during treatment. Additional CT scans are difficult to perform during the treatment for patients undergoing short-term treatments such as stereotactic body radiotherapy. If the improvement in the cone-beam CT images taken for setup verification is equivalent to the simulation CT images, additional CT scans can be eliminated.

4. While the methodology is comprehensive, a brief discussion on the evaluation of final segmentation models would enhance clarity. Detailing the criteria used for selecting the best-performing model among 50 epochs would ensure transparency and reproducibility.

Response

The explanation for the selection of the best-performance model was inadequate in the manuscript. Among the stored segmentation models, the model with the lowest Dice and cross-entropy loss on the validation dataset was selected as the best-performance model. We have added an explanation of this loss.

Before [Line 189 – 192]

For each epoch, the performance of the training model was evaluated with validation data, and the best-performing model among the 50 models from 50 epochs was selected as the final model for the patient; thus, 18 segmentation models were produced.

After [Line 191 – 193]

For 50 epochs, we saved the model at each epoch. Among the 50 models, the model with the lowest Dice and cross-entropy loss on the validation data was selected as the final model for the patient; thus, 18 segmentation models were produced.

5. The article needs major improvements in English language and there are many grammars and sentence error in this paper.

Response

The manuscript has been proofread by a professional English-language-editing agency.

Reviewer #2: 

This paper describes a technique of creating individualized patient neural networks for organ segmentation in patients receiving radiation therapy for prostate cancer. It is well conceived and executed and well written. Its main strength is the utilization of data augmentation to demonstrate good segmentation performance with very limited training data. The technical aspects are clearly described and many examples are clearly presented. Its main limitation is the limited number of patients (18) included.

A few specific comments/questions:

1. Were all patients who presented for radiation treatment at Samsung Medical Center between November 18, 2019 and September 24, 2021 included in the sample? If there were any patients excluded please explain why/exclusion criteria. Potential bias in such a small sample may limit generalizability of findings.

Response 

Not all patients were included in the sample. Only those with prostate cancer who agreed to undergo additional CT scans were included. Patients who did not consent to undergo additional CT scans were excluded.

Modified [Line 128 – 130]

We explained the research to adult male patients aged ≥18 years who underwent radiation treatment for prostate cancer between November 18, 2019, and September 24, 2021, and obtained signed consent forms from patients who expressed willingness to participate in additional CT scans.

2. In methods you state "if the loss decreased below 1000 and the generated image was similar to that of a human body, the DVFs were saved." Please clarify how "similar to that of a human body" was defined - was this by manual visual inspection? If so, what was the time required to do this?

Response

We agree that the sentence seems to be somewhat ambiguous. During the generation of the CT images using DVFs, when the loss exceeded 1000, the organs overlapped. When the loss dropped to below 1000, the three organs returned to their original positions. Therefore, when the loss of the augmentation method decreased to below 1000, the DVFs were saved and used as augmentation images. The time taken for the loss to decrease to below 1000 varied per patient and was generally approximately 3–10 min. This sentence has been revised accordingly.

Before [Line 163 – 166]

When the loss was large, 〖CT〗^moved generated via training did not appear to be similar to the image of the human body. After 250 iterations, which was equal to one epoch, if the loss decreased below 1000 and the generated image was similar to that of a human body, the DVFs were saved. As the iterations continued, nine more DVFs were saved such that ten DVFs were extracted for one image pair.

After [Line 163 – 168]

When the loss exceeded 1000, the organs in the generated 〖CT〗^moved overlapped, resulting in images that did not resemble the human anatomy. Therefore, we began saving the DVF as the loss decreased to below 1000. As the iterations continued, nine more DVFs were saved such that 10 DVFs were extracted for one image pair. Each epoch consisted of 250 iterations, and the training was terminated when 10 sets of DVFs with a loss below 1000 were completely saved. Approximately 3–10 min was required to save 10 sets of DVFs.

3. The individual models were compared with a total model containing data for just 18 patients. Is this a big enough "total model" or should further research be done to validate the individual model performance against a larger total model?

Response 

The data from 18 patients were not sufficiently large for a comprehensive model. As shown in Table 4, in other studies, an average of 300 datasets has typically been used, with up to 876 datasets being used for training. For the comparison of the individual and total models using our planned VoxelMorph augmentation method, obtaining patient data that include two datasets from the same patient was challenging. Although when different data were used, compared with other studies, our results showed similar performance with significantly fewer datasets. Thus, our results for only 18 datasets demonstrate the performance similar to a much larger dataset. If additional datasets are obtained to validate a larger total model, the appropriate number of datasets required can be determined.

4. In your abstract conclusion you state "The proposed method can be clinically adopted for automatic prostate segmentation for adaptive radiation therapy." Do you believe this model is ready for clinical use? How would you propose it be adopted - to generate a "first draft" radiation plan which would be reviewed by a physician prior to radiation being given to the patient?

Response

We demonstrated that the performance of the proposed method is equivalent to those of models trained with a large dataset. For clinical applicability in patient treatment, a clinical routine in which a 2nd CT is acquired needs to be established. As mentioned in the main text (lines348 -351), if we enhance the quality of daily CBCT images and use for our research method, auto segmentation for ART may be utilized clinically. In addition, any further performance improvement of the proposed model will be beneficial for clinical applications. Therefore, we have modified the abstract as follows:

modified [line 44 – 46, abstract conclusion]

We demonstrated the feasibility of automatic segmentation by employing individual patient datasets and image augmentation techniques. The proposed method has potential for clinical application in automatic prostate segmentation for ART.

5. Please proofread for minor errors, e.g. "Abstract" and peri

---

## [Decision Letter · Decision Letter 1]

18 Jul 2024

Deep-learning-based segmentation using individual patient data of prostate cancer radiation therapy.

PONE-D-23-40530R1

Dear Dr. Han,

We’re pleased to inform you that your manuscript has been judged scientifically suitable for publication and will be formally accepted for publication once it meets all outstanding technical requirements.

Kind regards,

Khan Bahadar Khan, Ph.D

Academic Editor

PLOS ONE

Additional Editor Comments (optional):

Reviewers' comments:

Reviewer's Responses to Questions

**Comments to the Author**

1. If the authors have adequately addressed your comments raised in a previous round of review and you feel that this manuscript is now acceptable for publication, you may indicate that here to bypass the “Comments to the Author” section, enter your conflict of interest statement in the “Confidential to Editor” section, and submit your "Accept" recommendation.

Reviewer #1: All comments have been addressed

Reviewer #2: All comments have been addressed

2. Is the manuscript technically sound, and do the data support the conclusions?

Reviewer #1: Yes

Reviewer #2: Yes

3. Has the statistical analysis been performed appropriately and rigorously? 

Reviewer #1: Yes

Reviewer #2: Yes

4. Have the authors made all data underlying the findings in their manuscript fully available?

Reviewer #1: Yes

Reviewer #2: No

5. Is the manuscript presented in an intelligible fashion and written in standard English?

Reviewer #1: Yes

Reviewer #2: Yes

6. Review Comments to the Author

Reviewer #1: The authors have incorporated all the requested changes in the revised manuscript, and it is now suitable for the next step with no further revisions needed.

Reviewer #2: (No Response)

7. PLOS authors have the option to publish the peer review history of their article (what does this mean?). If published, this will include your full peer review and any attached files.

Reviewer #1: No

Reviewer #2: No

---

## [Editor Report · Acceptance letter]

23 Jul 2024

PONE-D-23-40530R1 

PLOS ONE

Dear Dr. Han, 

I'm pleased to inform you that your manuscript has been deemed suitable for publication in PLOS ONE. Congratulations! Your manuscript is now being handed over to our production team.

Kind regards, 

on behalf of

Dr. Khan Bahadar Khan 

Academic Editor

PLOS ONE